# Early Skin Test after Anaphylaxis during Induction of Anesthesia: A Case Report

**DOI:** 10.3390/medicina56080394

**Published:** 2020-08-07

**Authors:** Ann Hee You, Jeong Eun Kim, Taewan Kwon, Tae Jun Hwang, Jeong-Hyun Choi

**Affiliations:** Department of Anesthesiology and Pain Medicine, Kyung Hee University Hospital, Kyungheedae Road 23, Dongdaemun-Gu, Seoul 02447, Korea; annhee.you@gmail.com (A.H.Y.); promemoria27@hanmail.net (J.E.K.); iris833@naver.com (T.K.); toranous@naver.com (T.J.H.)

**Keywords:** anaphylaxis, skin prick test, rocuronium, cisatracurium, general anesthesia

## Abstract

Background: It is recommended that a skin test be performed 4–6 weeks after anaphylaxis. However, there is little evidence about the timing of the skin test when there is a need to identify the cause within 4–6 weeks. Case report: A 57-year-old woman was scheduled to undergo surgery via a sphenoidal approach to remove a pituitary macroadenoma. Immediately after the administration of rocuronium, pulse rate increased to 120 beats/min and blood pressure dropped to 77/36 mmHg. At the same time, generalized urticaria and tongue edema were observed. Epinephrine was administered and the surgery was postponed. Reoperation was planned two weeks after the event. Four days after the anaphylactic episode, rocuronium was confirmed to be the cause by the skin prick test. Cisatracurium, which showed a negative reaction, was selected as an alternative agent for future procedures. Two weeks later, the patient underwent reoperation without any adverse events. Conclusions: The early skin test can be performed if there is a need even earlier than 4–6 weeks after anaphylaxis.

## 1. Introduction

Anaphylaxis is estimated to occur in one per 7000 to 10,000 cases of anesthesia, and neuromuscular blocking agents and antibiotics are the most common causes [1,2]. When anaphylaxis occurs in general anesthesia, it is necessary to identify the causative agent and choose an alternative agent for future procedures. The gold standard for the diagnosis of anaphylaxis is a drug provocation test that has a risk of life-threatening complications. According to the diagnostic algorithm for perioperative hypersensitivity by Ebo et al., skin prick tests with in vitro tests are used for diagnosis [3]. However, it is recommended the skin test be performed 4–6 weeks after anaphylaxis. Many studies about in vitro tests have been published, however, there is no guideline if there is a need for early surgical reintervention. 

The patient in this case whose surgery was delayed due to anaphylaxis during induction of general anesthesia was able to undergo an early reoperation in two weeks through an early skin test. The purpose of this report is to emphasize the necessity of early skin tests and to review the relevant literature.

## 2. Case Presentation

A 57-year-old woman was diagnosed with a pituitary macroadenoma and scheduled to undergo surgery via a sphenoidal approach to remove the tumor. She had a history of urticaria caused by pollen and Computed Tomography contrast agent. There were no abnormal findings on preoperative physical examination, blood tests, chest radiography, and electrocardiography.

Noninvasive blood pressure (BP), heart rate (HR), electrocardiogram, and peripheral oxygen saturation (SpO2) were monitored as the patient entered the operation theater. The initial vital signs were BP, 168/81 mmHg; HR, 82 beats/min; SpO2, 98%. Preoxygenation was performed and 0.2 mg of glycopyrrolate was administered as premedication. Induction was performed with propofol and remifentanil via a target-controlled infusion pump. When the patient lost consciousness, 60 mg of rocuronium (Esmeron, N.V., Organon, Oss, The Netherlands) was injected. Immediately after administration, HR increased to 120 beats/min and BP dropped to 77/36 mmHg. At the same time, generalized urticaria and tongue edema were observed (Figure 1A,B). Despite the administration of phenylephrine and fluid, BP continued to drop to 57/34 mmHg and manual mask ventilation became difficult as resistance increased. Intubation was performed immediately, and a radial artery catheter was inserted for continuous monitoring of BP. We suspected the occurrence of anaphylaxis and administered 20μg of epinephrine intravenously. The hemodynamics were stabilized with systolic BP maintained between 110 and 150 mmHg. All the agents used during the induction of anesthesia were stopped. To relieve symptoms of anaphylaxis, 4 mg of chlorpheniramine and 100 mg of hydrocortisone were administered. According to the modified Ring and Messmer four-step grading scale, the patient was classified as grade III [4]. After discussion with the neurosurgeon, we decided to postpone the operation. In total, 4 mg/kg of sugammadex was administered for the reversal of muscle relaxation. After confirming the recovery of consciousness and spontaneous breathing, extubation was performed.

The patient was transferred to the ward after her vital signs and respiration had stabilized in the post-anesthesia care unit. A serum tryptase test was performed 6 h after the anaphylaxis, and the result was within the normal range of 6.4 μg/L.

Four days after the episode, we referred our patient to a dermatology department for a skin prick test. The concentration of the agents followed the Australian and New Zealand Anesthetic Allergy Group (ANAZAAG) perioperative anaphylaxis investigation guideline [5]. Most of the agents were used undiluted except palonosetron and iopamidol. In total, 0.15 mg/mL of palonosetron and 755 mg/mL of iopamidol were diluted to 1:100. A total of 10 mg/mL of rocuronium, 10 mg/mL of propofol, 0.05 mg/mL of remifentanil, 2 mg/mL of cisatracurium, and 4 mg/mL of vecuronium were used for the test. An amount of 0.9% normal saline (1) was used for the negative control and 0.1% histamine (10) was used for the positive control. We enlisted some identical drugs but from different manufacturers containing different preservatives and additives. Table 1 shows the result of the test. The responses were confirmed at 5 and 30 min (Figure 2A,B). Wheals and flares were judged by a dermatologist. The wheals and flares were observed in 5 and 30 min in rocuronium and histamine, which were used as controls. Rocuronium was suspected to be the agent that caused the anaphylaxis. Cisatracurium and vecuronium, were included in the test for identifying an alternative neuromuscular blocking agent for reoperation, which appeared to be negative.

The intradermal test was not done because the skin test was positive. The surgery was re-planned two weeks after the event. Anesthesia was induced and maintained with propofol and remifentanil. After loss of consciousness, 10 mg of cisatracurium was administered for muscle relaxation. The patient’s vital signs were stable. The patient underwent surgery without any other adverse event and was discharged without any complications 7 days later. Written informed consent was obtained from the patient for the publication of this case report.

## 3. Discussion

The range of anaphylaxis and perioperative hypersensitivity has been expanded to incorporate non-immune-mediated life-threatening, generalized, and systemic reactions into it [6]. Symptoms of anaphylaxis vary and may include hypotension, tachycardia, cardiovascular collapse, angioedema, bronchospasm, wheezing, hypoxia, pulmonary edema, and urticaria. In a new clinical scoring system, the impact of confounding factors and timing are also calculated [7]. Diagnosis of anaphylaxis which occurs in general anesthesia is challenging due to the concurrent administration of multiple drugs within a short period of time. Moreover, most anesthetic agents cause hypotension secondary to vasodilatation. When skin symptoms are accompanied by bronchospasm or hypotension, and hypotension that does not respond to vasopressor agents appears, the observers should suspect anaphylaxis.

In this case, we performed an early skin test to identify the cause of anaphylaxis which occurred during induction of anesthesia. After confirming the positive reaction for rocuronium, cisatracurium which showed a negative reaction was selected as an alternative neuromuscular blocking agent, and the surgery was performed safely two weeks later.

Through this case, we found that an early skin test provides useful clinical information on the causative agent if the patient requires early reoperation. The general recommendation is to perform the skin test 4–6 weeks after anaphylaxis. This is due to the possibility of false negatives caused by the exhaustion of histamine from mast cells or basophils. However, it is difficult to apply the same criteria to identify the causative agent in the perioperative period for patients in need of early reoperation. 

According to Takazawa et al., the skin prick test has low sensitivity when reintervention is needed in four weeks. They suggested specific IgE quantification for identifying the causative agent [8]. However, specific IgE tests showed a very low specificity due to nonspecific binding, especially in rocuronium anaphylaxis [9,10]. Other in vitro tests including tryptase, MPGPRX2, basophil, and histamine are also active, but all have some weaknesses [3,11,12].

As peak tryptase level indicates mast cell degranulation, >2 + 1.2 × baseline tryptase is thought to be clinically relevant [12]. The threshold of tryptase was set at 11.4 to 25 µg/L, but comparing it with baseline values has become more widely used [13]. However, the baseline tryptase is not one of the routine laboratory tests in many centers, and the prediction of anaphylaxis is mostly impossible. Timed sampling for tryptase is difficult in many cases. Additionally, the sensitivity of the test varies in the literature, and some patients presented anaphylaxis clinically without elevated tryptase level [8].

According to the guidelines of the British Society for Allergy and Clinical Immunology (2010), there are no reasons to delay the skin test and the test can be performed immediately after recovering from the anaphylactic reaction [14]. The guidelines from ANAZAAG state that the skin test should ideally be performed 4–6 weeks after the anaphylactic reaction except for some cases [5]. Even though it is not mandatory to delay the skin test, there are limited cases or reports to guide the test when the patient requires early intervention.

In 2016, Schulberg EM et al. [15] reported a similar case as in this report of an early skin test after anaphylaxis due to rocuronium. They also observed a negative skin test for cisatracurium; however, the surgery was completed without the administration of the neuromuscular blocking agent. In our case, the surgery was safely performed using cisatracurium, which was selected as the alternative agent [15].

There are some limitations to this study. In our case, we performed the surgery safely using cisatracurium, which was selected as the alternative agent in an early skin test. However, we should be careful in the administration of the agent even if the result is negative, because it could change to positive later [15,16]. In addition, tryptase sampling is recommended three times—1, 2–4, and 24 h after the anaphylaxis reaction [2]. In this case, the tryptase was examined only once, 6 h after the event, and the result was within the normal range. The half-life of tryptase is 2 h and delay in performing the test may have affected the result. According to Garvey et al., latex and chlorhexidine should have been tested regardless of exposure [17]. However, as the quality of latex has improved and gloves without powder are being widely used, the incidence of anaphylaxis due to latex has considerably decreased. Incidence due to chlorhexidine varies from 1.0 to 9.6% depending on the study [12].

There are no consensus or guidelines for the timing of skin test for patients who require early reoperation. Through this case, we found that an early skin test provides useful clinical information on the causative agent. The early skin test can be performed if there is a need even within 4–6 weeks after anaphylaxis.

## 4. Conclusions

If there is a need for general anesthesia within 4–6 weeks after anaphylaxis during induction, an early skin test can offer clinical information that could be used in the next operation. However, the possibility of false negatives cannot be ignored and thus, anesthesiologists have to be prepared for a recurrence of anaphylaxis during the skin test or reoperation.

## Figures and Tables

**Figure 1 medicina-56-00394-f001:**
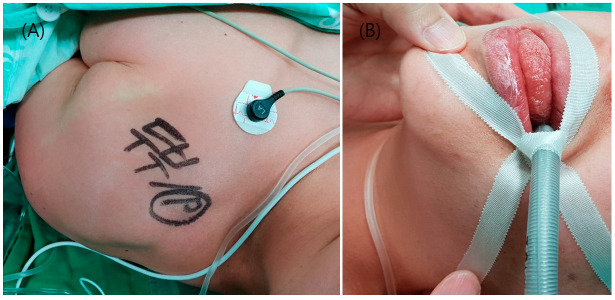
Generalized urticaria (**A**) and tongue edema (**B**) due to anaphylaxis.

**Figure 2 medicina-56-00394-f002:**
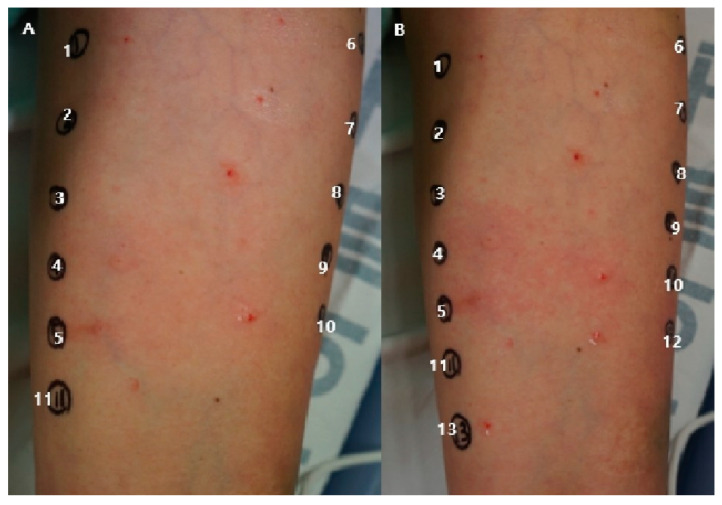
Results of the skin prick test in 5 (**A**) and 30 min (**B**).

**Table 1 medicina-56-00394-t001:** Results of skin prick test four days after anaphylaxis.

	Substances	Response			
		5 min		30 min	
		Wheal	Flare	Wheal	Flare
1	0.9% Normal saline (negative control)	-	-	-	-
2	Plasma solution	-	-	-	-
3	Palonosetron (Palseron, Samyang, Korea)	-	-	-	-
4	Rocuronium (Rocuron, Myungmoon, Korea)	1+	1+	1+	1+
5	Rocuronium (Esmeron, Organon, Netherlands)	1+	1+	1+	1+
6	Propofol (Fresofol 2%^TM^, Fresenius Kabi, Austria)	-	-	-	-
7	Propofol (Pofol, Dongkook, Korea)	-	-	-	-
8	Remifentanil (Tivare, BCWORLD, Korea)	-	-	-	-
9	Remifentanil (Ultian, Hanlim, Korea)	-	-	-	-
10	Histamine (positive control)	1+	1+	1+	1+
11	Iopamidol (Pamiray, Dongkook, Korea)	-	-	-	-
12	Cisatracurium (Nimbex, Mitsubishi Tanabe, Japan)	-	-	-	-
13	Vecuronium (Vecuron, Myungmoon, Korea)	-	-	-	-

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
