# Peer review of "Early Skin Test after Anaphylaxis during Induction of Anesthesia: A Case Report"

_medicina, 2020, doi:10.3390/medicina56080394_

Round 1
Reviewer 1 Report
Thank you for the opportunity to revise this case report.
-The introduction should be improved. The problem was reported clearly, however the authors missed to explain what is the problem related to the timing. The authors stated that the timing is not clear however, it is not clear the reason why there are limited evidences. Please improve introduction section
-What about patient' consent to publish this case report and figures?
-in case description, the authors should describe the laboratory test for the early skin test
- I suggest improving the discussion section. The authors should report in detail the current literature evidence and explain the clinical implication of using early skin test
Author Response
Thank you for your kind comments.
-The introduction should be improved. The problem was reported clearly, however the authors missed to explain what is the problem related to the timing. The authors stated that the timing is not clear however, it is not clear the reason why there are limited evidences. Please improve introduction section
The introduction was revised focused on ‘timing’ using up to dated literatures.
-What about patient' consent to publish this case report and figures?
The sentence “Written informed consent was obtained from the patient for the publication of this case report.” was inserted in the last part of case presentation and an informed consent was sent to the editor on July 13th.
-in case description, the authors should describe the laboratory test for the early skin test
We described the early skin prick test in the case presentation. The guideline we referred and the concentrations of the agents are inserted and table 1 and figure 2 was revised for clarity.
- I suggest improving the discussion section. The authors should report in detail the current literature evidence and explain the clinical implication of using early skin test
The discussion section was improved using current literatures.
Reviewer 2 Report
Early skin test after anaphylaxis during induction of 3 anesthesia: A case report by Ann Hee You
To this reviewer it appears that both the case report and the discussion need major revision / improvement. With respect to the case report essential data is missing. E.g. was there any evidence for mast cell activation? Which concentrations for skin prick testing were used (all compounds!)? Was intradermal testing (IDT) offered when skin prick tests were negative (imperative as a negative skin prick test is not conclusive, even negative IDT cannot always provide green light!)? The figures are unclear and do not really add to the manuscript (wheals and flares should be denoted in the table for all compounds).
Lines 82-88 are superfluous.
The discussion: some references are long outdated. The authors are urged to update their literature. There is a very interesting issue of the British Journal of Anesthesia 2019 with many papers relevant for this case report. The same holds true for the EAACI recommendations published in Allergy recently (Garvey L et al) and the numerous papers about rocuronium allergy published by the group of Ebo D et al. Actualizing the references and amending the discussion accordingly should improve the manuscript considerably.
Some recent examples:
1: Van Der Poorten MM, Molina-Molina G, Van Gasse AL, Hagendorens MM, Faber MA, De Puysseleyr L, Elst J, Mertens CM, Horiuchi T, Sabato V, Ebo DG. Application of specific-to-total IgE ratio does not benefit diagnostic performance of serologic testing for rocuronium allergy. Br J Anaesth. 2020 Jun 30:S0007-0912(20)30414-1. doi: 10.1016/j.bja.2020.05.032. Epub ahead of print. PMID: 32620258.
2: Garvey LH, Melchiors BB, Ebo DG, Mertes PM, Krøigaard M. Medical algorithms: Diagnosis and investigation of perioperative immediate hypersensitivity reactions. Allergy. 2020 Feb 13. doi: 10.1111/all.14226. Epub ahead of print. PMID: 32053736.
3: Garvey LH, Dewachter P, Hepner DL, Mertes PM, Voltolini S, Clarke R, Cooke P, Garcez T, Guttormsen AB, Ebo DG, Hopkins PM, Khan DA, Kopac P, Krøigaard M, Laguna JJ, Marshall S, Platt P, Rose M, Sabato V, Sadleir P, Savic L, Savic S, Scherer K, Takazawa T, Volcheck GW, Kolawole H. Management of suspected immediate perioperative allergic reactions: an international overview and consensus recommendations. Br J Anaesth. 2019 Jul;123(1):e50-e64. doi:10.1016/j.bja.2019.04.044. Epub 2019 May 24. PMID: 31130272.
4: Hopkins PM, Cooke PJ, Clarke RC, Guttormsen AB, Platt PR, Dewachter P, Ebo DG, Garcez T, Garvey LH, Hepner DL, Khan DA, Kolawole H, Kopac P, Krøigaard M, Laguna JJ, Marshall SD, Mertes PM, Rose MA, Sabato V, Savic LC, Savic S, Takazawa T, Volcheck GW, Voltolini S, Sadleir PHM. Consensus clinical scoring for suspected perioperative immediate hypersensitivity reactions. Br J Anaesth. 2019 Jul;123(1):e29-e37. doi: 10.1016/j.bja.2019.02.029. Epub 2019 Apr 24. PMID: 31029409.
5: Garvey LH, Ebo DG, Mertes PM, Dewachter P, Garcez T, Kopac P, Laguna JJ, Chiriac AM, Terreehorst I, Voltolini S, Scherer K. An EAACI position paper on the investigation of perioperative immediate hypersensitivity reactions. Allergy. 2019 Oct;74(10):1872-1884. doi: 10.1111/all.13820. Epub 2019 Jun 18. PMID: 30964555.
6: Ebo DG, Clarke RC, Mertes PM, Platt PR, Sabato V, Sadleir PHM. Molecular mechanisms and pathophysiology of perioperative hypersensitivity and anaphylaxis: a narrative review. Br J Anaesth. 2019 Jul;123(1):e38-e49. doi: 10.1016/j.bja.2019.01.031. Epub 2019 Mar 8. PMID: 30916022.
7: Mertes PM, Ebo DG, Garcez T, Rose M, Sabato V, Takazawa T, Cooke PJ, Clarke RC, Dewachter P, Garvey LH, Guttormsen AB, Hepner DL, Hopkins PM, Khan DA, Kolawole H, Kopac P, Krøigaard M, Laguna JJ, Marshall SD, Platt PR, Sadleir PHM, Savic LC, Savic S, Volcheck GW, Voltolini S. Comparative epidemiology of suspected perioperative hypersensitivity reactions. Br J Anaesth. 2019 Jul;123(1):e16-e28. doi: 10.1016/j.bja.2019.01.027. Epub 2019 Mar 4. PMID: 30916015.
8: Takazawa T, Sabato V, Ebo DG. In vitro diagnostic tests for perioperative hypersensitivity, a narrative review: potential, limitations, and perspectives. Br J Anaesth. 2019 Jul;123(1):e117-e125. doi: 10.1016/j.bja.2019.01.002. Epub 2019 Feb 12. PMID: 30915999.
9: Ebo DG, Van Gasse AL, Decuyper II, Uyttebroek A, Sermeus LA, Elst J, Bridts CH, Mertens CM, Faber MA, Hagendorens MM, De Clerck LS, Sabato V. Acute Management, Diagnosis, and Follow-Up of Suspected Perioperative Hypersensitivity Reactions in Flanders 2001-2018. J Allergy Clin Immunol Pract. 2019 Sep-Oct;7(7):2194-2204.e7. doi: 10.1016/j.jaip.2019.02.031. Epub 2019 Mar 8. PMID: 30857939.
10: Van Gasse AL, Elst J, Bridts CH, Mertens C, Faber M, Hagendorens MM, De Clerck LS, Sabato V, Ebo DG. Rocuronium Hypersensitivity: Does Off-Target Occupation of the MRGPRX2 Receptor Play a Role? J Allergy Clin Immunol Pract. 2019 Mar;7(3):998-1003. doi: 10.1016/j.jaip.2018.09.034. Epub 2018 Oct 10. PMID: 30315997.
11: Sabato V, Ebo DG. Hypersensitivity to Neuromuscular Blocking Agents: Can Skin Tests Give the Green Light for Re-Exposure? J Allergy Clin Immunol Pract. 2018 Sep-Oct;6(5):1690-1691. doi: 10.1016/j.jaip.2018.02.008. PMID: 30197072.
12: Ebo DG, Faber M, Elst J, Van Gasse AL, Bridts CH, Mertens C, De Clerck LS, Hagendorens MM, Sabato V. In Vitro Diagnosis of Immediate Drug Hypersensitivity During Anesthesia: A Review of the Literature. J Allergy Clin Immunol Pract. 2018 Jul-Aug;6(4):1176-1184. doi: 0.1016/j.jaip.2018.01.004. Epub 2018 Feb 14. PMID: 29454709.
Author Response
Thank you for your precious review.
To this reviewer it appears that both the case report and the discussion need major revision / improvement. With respect to the case report essential data is missing. E.g. was there any evidence for mast cell activation? Which concentrations for skin prick testing were used (all compounds!)? Was intradermal testing (IDT) offered when skin prick tests were negative (imperative as a negative skin prick test is not conclusive, even negative IDT cannot always provide green light!)? The figures are unclear and do not really add to the manuscript (wheals and flares should be denoted in the table for all compounds).
Mast cell activation- The patient did not show elevation in tryptase concentration. However, there are patients who presented anaphylaxis clinically without elevated tryptase level. The timing of the sampling could be a confounding factor.
Figure 2 is changed for visibility. The figure of right after the test was deleted and shadowy numbers were clearly marked. The indications of the numbers were listed and comments added.
In table 1, we added the product name and manufacturers. Instead of using the word ‘CT contrast agent’ in 11, we substitute it to iopamidol. The expression of the result was changed. Wheals and flares are presented.
Case presentation-We described the skin prick test in detail including the concentrations of the agents. Line 78 ‘The intradermal test was not offered because the skin test was clearly positive.’ was inserted.
Lines 82-88 are superfluous. We deleted lines 82-88 for clarity.
The discussion: some references are long outdated. The authors are urged to update their literature. There is a very interesting issue of the British Journal of Anesthesia 2019 with many papers relevant for this case report. The same holds true for the EAACI recommendations published in Allergy recently (Garvey L et al) and the numerous papers about rocuronium allergy published by the group of Ebo D et al. Actualizing the references and amending the discussion accordingly should improve the manuscript considerably.
We revised the discussion section. Lists of updated literatures were really helpful to revise our manuscript. We updated some references using the lists.
Round 2
Reviewer 1 Report
No comments